# Monitoring and Analysis of Water Level–Water Storage Capacity Changes in Ngoring Lake Based on Multisource Remote Sensing Data

**Weidong Zhu** [1,2,3] , **Shubing Zhao** [1,] * , **Zhenge Qiu** [1,2] , **Naiying He** [1,2] , **Yaqin Li** [1] , **Ziya Zou** [1] **and Fei Yang** [1]

1 College of Marine Sciences, Shanghai Ocean University, Shanghai 201306, China; wdzhu@shou.edu.cn (W.Z.); zgqiu@shou.edu.cn (Z.Q.); nyhe@shou.edu.cn (N.H.); m210200572@st.shou.edu.cn (Y.L.); m210200585@st.shou.edu.cn (Z.Z.); ywli@shou.edu.cn (F.Y.)
2 Estuary Marine Surveying and Mapping Engineering Technology Research Center, Shanghai 201306, China
3 Key Laboratory of Marine Ecological Monitoring and Restoration Technologies, Shanghai 201306, China
* Correspondence: m200200655@st.shou.edu.cn

**Abstract:** Mastering the fluctuation of water levels and the water storage capacity of plateau lakes is greatly important for monitoring the water balance of the Tibetan Plateau and predicting regional and global climate change. The water level of plateau lakes is difficult to measure, and the ground measured data of long-time series are difficult to obtain. Ngoring Lake is considered in this study, using spaceborne single-photon lidar ICESat-2/ATL13 inland lake standard data products, the water level values provided by Hydroweb laboratory, and the image data of an optical remote sensing satellite. A new method is proposed in the absence of measured data. The method uses multisource remote sensing data to estimate the long-term changes in the water levels, surface area, and water storage capacity of Ngoring Lake in the past three decades. The results show that the water level values of ICESat-2 and Hydroweb on overlapping observation days are highly correlated, with $R^2 = 0.9776$, $MAE = 0.420$ m, $RMSE = 0.077$ m, and the average absolute height difference is 0.049 m. The fusion of multiple altimetry data can obtain more continuous long-time series water-level observation results. From 1992 to 2021, the water body information of Ngoring Lake basin fluctuated greatly and showed different variation characteristics in different time periods. The lowest water level in January 1997 was approximately 4268.49 m, and it rose to its highest in October 2009, approximately 4272.44 m. The change in the water level in the basin was mainly affected by natural factors, such as precipitation, air temperature, and human activities. The analysis shows that ICESat-2 can be combined with other remote sensing data to realize the long-time series dynamic monitoring of plateau lakes, showing great advantages in the comprehensive observation of plateau lakes in no man's land.

**Keywords:** lake water level; water storage; ICESat-2; remote sensing image; Ngoring Lake

## 1. Introduction

Water is a finite, vulnerable and essential resource which should be managed in an integrated manner [1]. Lakes are an important carrier of water resources, which are crucial for maintaining the balance of watershed ecosystems, regional/global climate stability, and socio-economic development [2]. The water level is among the most important characteristics of lakes. Real-time monitoring by establishing long-term water level sequences helps to estimate the fluctuation of lake water reserves. This enables one to understand regional climate change so as to formulate appropriate environmental protection policies and realize the benign development of regional ecological environment [3].

The Tibetan Plateau (TP) is a sensitive and significant area of global climate change, with an average altitude of over 4000 m. Lakes are the main landscape elements occupying the TP. The region has a well-developed water system, with more than 1500 lakes, among

which more than 900 lakes cover an area of more than 1 km². Therefore, it is called the "Third Pole" and "Asian Water Tower" [4]. Especially on the TP, extreme precipitation [5] is a valuable contributor to water resources and plays a key role in the inland water cycle [6–8]. However, the number of traditional water level monitoring stations is scarce, the monitoring time is short, and the continuity of data observation is poor due to the harsh climatic environment, complex terrain, and traffic occlusion on the plateau. Alpine hypoxia and backward equipment technology make manual field measurement more difficult. To a certain extent, this condition has become a bottleneck restricting the sustainable utilization of regional water resources and the analysis of the law of climate change, especially the research on the response mechanism of the plateau region to global climate change [9]. Ngoring Lake is a large plateau freshwater lake at the source of the Yellow River. It regulates the runoff of major rivers in the middle and lower reaches of the Yellow River and plays an important role in maintaining ecological balance [10]. Therefore, it is of great significance and value to study the water level–storage variation of Ngoring Lake at the source of the Yellow River.

With the rapid development of high-resolution optical remote sensing satellites and laser altimeter satellite technologies, new research approaches have been provided for the long-term monitoring of lake water levels and surface areas in plateau lakes. The water-level inversion based on satellite remote sensing technology has experienced a transition from optical to microwave (spaceborne altimeter), single to multisource joint inversion [11]. Optical remote sensing detection is limited by digital elevation model (DEM) accuracy, atmospheric conditions, cloud layer thickness, and imaging time; it has certain limitations in practical applications [12]. Satellite altimetry technology has the characteristics of globality, periodicity, and a strong penetrating ability which can quickly and accurately obtain water level information, detect various natural phenomena and changes in surface water. It has more prominent advantages in plateau areas and areas with frequent natural disasters [13]. Combining the two techniques for inland water monitoring cannot only compensate for the limitations of a single data source, but also obtain more accurate observations [14].

In recent years, remote sensing technology has been used to monitor inland water bodies, and many related studies have been carried out at home and abroad. Jiang et al. [15] calculated the water level change of Qinghai Lake by using ENVISAT data and based on waveform retracing technology, which was verified by the measured data of water level station and achieved good monitoring results. Adalbert Arsen et al. [16], combined with MODIS Images and ICESat waveform data, conducted an inversion study of the lake water level and water storage capacity from 2000 to 2012 for Lake Poopó, which has no data records to date. This method has certain applicability to similar sparse vegetation areas and arid areas. A.K. Dubey et al. [17] inverted the water level of the Yarlung Zangbo River basin based on the ICESat waveform retracking algorithm, and the root mean square error was 50–55 cm. This method effectively improves the accuracy of water level calculation by altimeter in different reaches of Yarlung Zangbo River. Based on remote sensing images, satellite altimetry data, and hydrometeorological parameters, Frederick et al. [18] used datasets for multiple regression analysis to establish a prediction model of the total surface water area of Lake Chad. The correlation between the total surface water area in a given month and one or more other hydrological parameters was analyzed and verified. The results show that the average absolute error of the regression equation in the LOOCV test is between 5.3% and 7.6%, which is better than the prediction made using the average value of the total surface water area in a given month or last year. Wang et al. [19] used the GEE geospatial analysis platform to analyze the Landsat series of satellite images from 1989 to 2019 to obtain the annual surface area time series data of 976 lakes with the largest area exceeding 1 km² in the inland basin of the TP. They also analyzed the relationship between the lake area and climate variables. These studies provide accurate estimates for the changes in lake water level and surface area. The research results are greatly important for water resource management under the background of climate change.

In summary, most current studies have mainly focused on the changes in the lake water level and surface area, while studies on the fluctuation of lake water storage capacity are limited [20,21]. The water storage capacity of most lakes on the TP is unknown due to the lack of bathymetric data to estimate the water storage [21]. The change in lake water storage capacity is an essential response of lakes to climate change. It has a better response ability than the change in lake area caused by different terrain conditions [22]. In the present study, Ngoring Lake in Qinghai Province is considered an example, and multisource altimetry data are applied to lake water level monitoring. Combined with Landsat series and Sentinel-2 satellite images, the functional relationship between the water level and water storage capacity is constructed, and Ngoring Lake has been continuously monitored for nearly 30 years, including the changes in water level, lake area, and water storage capacity. Compared with previous studies [16–18], the data and methods used in this paper are novel and have a longer observation time. It is of great benefit to the follow-up study of lakes.

## 2. Materials and Methods

### 2.1. Study Area

Ngoring Lake is located in the west of Maduo County, Qinghai Province. The geographical range is 34°46′03″ N–35°05′21″ N; 97°29′21″ E–97°54′21″ E (Figure 1). It is one of the largest freshwater lakes on the TP [10]. The North–South length of Ngoring Lake is approximately 32.3 km, and its East–West width is approximately 31.6 km. The lake surface is 4272 m above sea level, and the average water depth is 17.6 m. The deepest point north of the lake center can reach 32 m [23]. The upper source of the Yellow River flows in from the southwest of Ngoring Lake and flows out from the northeast. The sediment concentration of the lake is low, and the lake is bluish blue. The main supply sources of the lake water are surface runoff and natural precipitation, with an average annual precipitation of 321.6 mm [24].

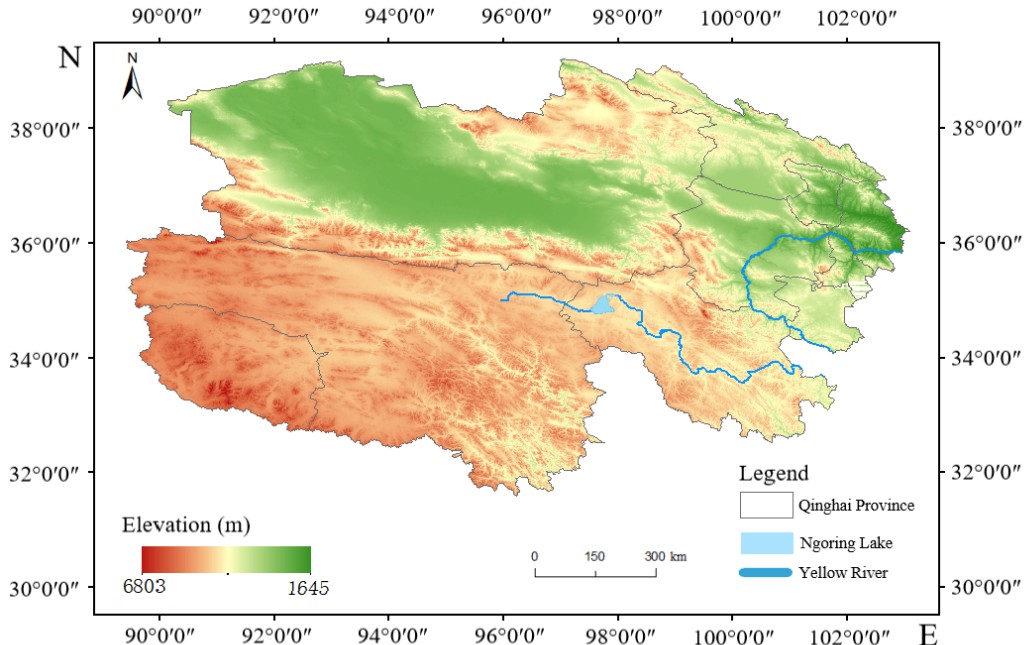

**Figure 1.** Geographical Location and Scope of Ngoring Lake in Qinghai Province.

According to the observation data of Maduo County Meteorological Station affiliated to Ngoring Lake, the annual average temperature in the Ngoring Lake basin is approximately −4.6 °C, and the annual precipitation is approximately 200–400 mm [25]. The hottest months are July and August, and the monthly average temperature is approximately 8 °C,

but the daily minimum temperature still drops below 0 °C. The coldest month is January, and the monthly average temperature is approximately −16.9 °C. The freezing period starts in the middle and late November, with an average of 157 days; the longest annual freezing period is approximately 193 days, and the freezing period is more than half a year [26]. The lakeside vegetation is a subalpine meadow, and the soil types are cold desert soil, meadow soil, chernozem soil, marsh soil, saline soil, and cinnamon soil. Grassland degradation and soil desertification are serious due to overgrazing and climate change [24].

### 2.2. Data

### 2.2.1. ICESat-2 Laser Altimeter Data

ICESat-2 (Ice, Cloud, and land Elevation Satellite-2) is a new generation of laser altimeter satellite with a photon counting system. Its main load is the Advanced Topographic Laser Altimeter System (ATLAS). It was successfully launched in the United States on 15 September 2018 [27]. The observation coverage of ICESat-2 is 88° S~88° N, and the revisit period is 91 days. As shown in Figure 2, ATLAS transmits laser pulses (532 nm) at a frequency of 10 k Hz, with a pulse width of 1.5 ns, and can obtain overlapping spots with an interval of approximately 0.7 m and a diameter of approximately 17 m along the track [28]. The laser pulse is divided into six beams by the diffractive optical elements in ATLAS, and three pairs are arranged along the track direction. Each pair consists of a strong beam and a weak beam, and the energy ratio of the two is approximately 4:1 [29]. ICESat-2 has 21 types of standard data products. In this study, the 3A-level data product ATL13 is adopted; it mainly provides inland and near-coastal water level elevation values along the track [30]. Since September 2018, the ATL13 data actually available in Ngoring Lake basin include 29 days in total. The location distribution and date information of the observed data are shown in Figure 3 and Table 1, respectively.

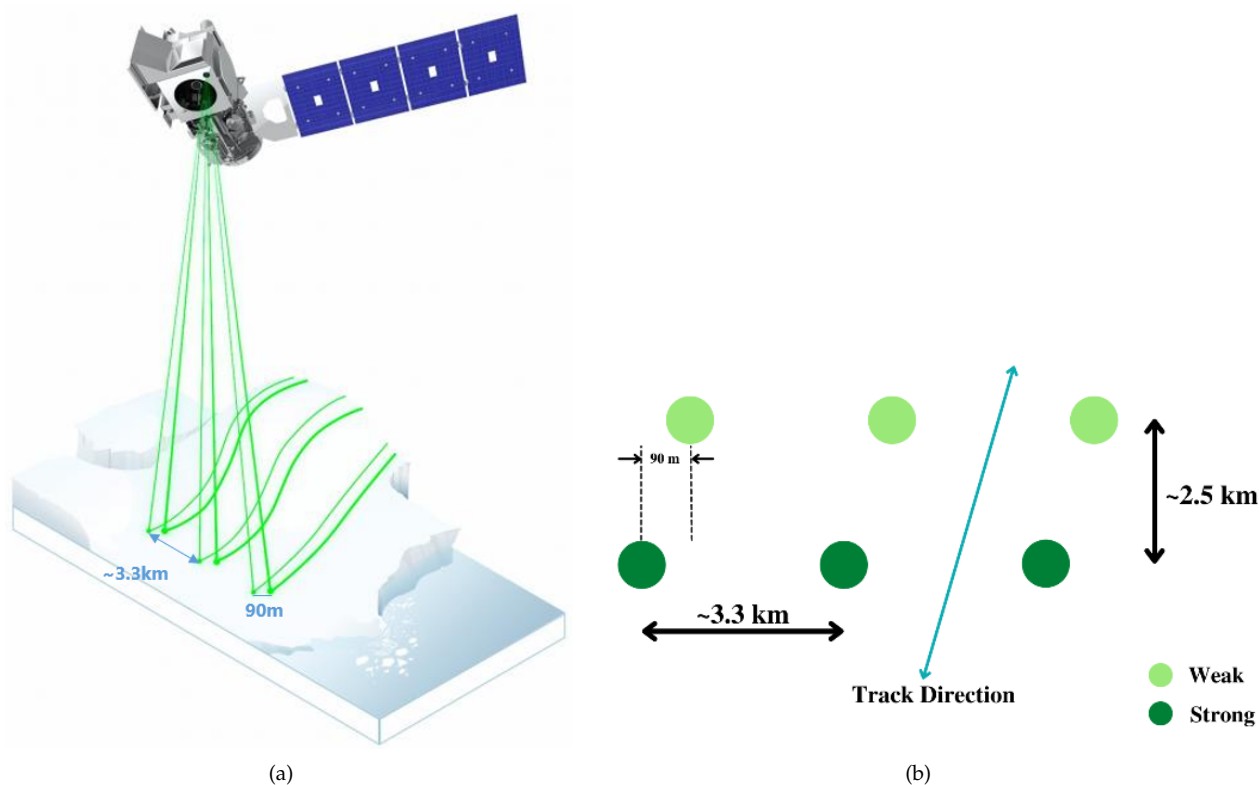

(a)

(b)

**Figure 2.** ICESat-2/ATLAS beams. (**a**) Schematic diagram of the operation of ICESat-2; (**b**) Flight orbit and photon diagram.

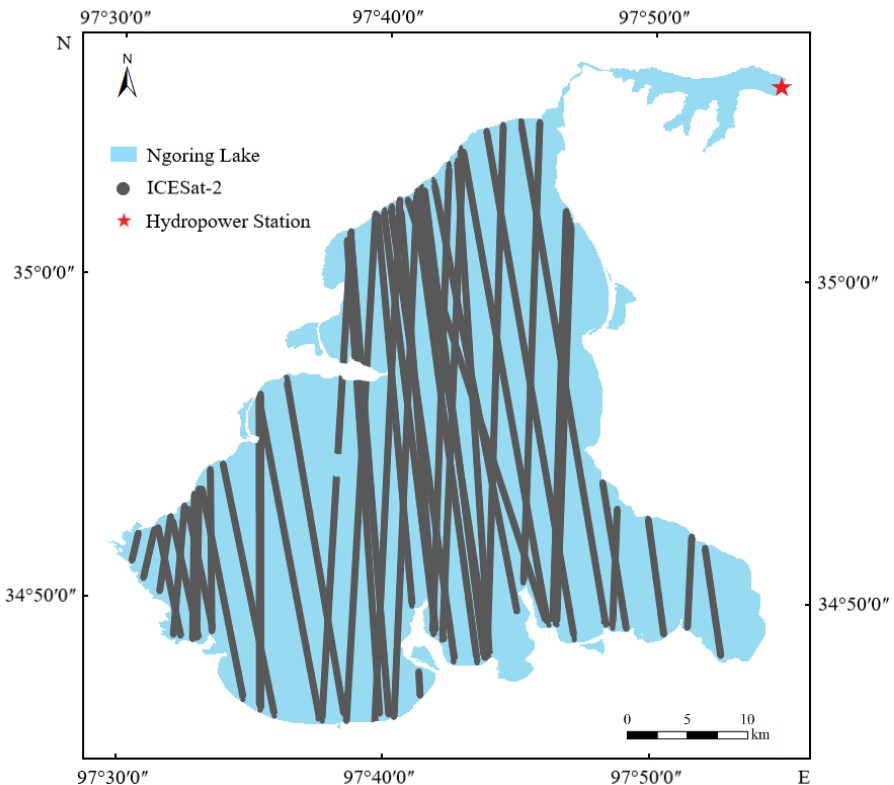

**Figure 3.** Distribution of ICESat-2 data across Ngoring Lake.

**Table 1.** Summary of ICESat-2/ATL13 Data across Ngoring Lake.

| Track Number | Observation Date |
| --- | --- |
| ID 248 | 14/10/2018; 14/07/2019; 11/01/2020; 11/07/2020; 10/10/2020; 10/04/2021; |
| ID 393 | 23/07/2019; 21/01/2020; |
| ID 690 | 12/08/2019; 07/02/2021; |
| ID 835 | 22/11/2018; 21/02/2019; 22/05/2019; 20/11/2019; 19/02/2020; 20/05/2020; 19/08/2020; 17/11/2020; 16/02/2021; |
| ID 1193 | 15/12/2018; 16/03/2019; 15/06/2019; 12/06/2020; 11/12/2020; 10/06/2021; |
| ID 1277 | 22/03/2019; 19/09/2019; 18/06/2020; 16/12/2020; |

2.2.2. Optical Remote Sensing Image Data

The remote sensing image data used in Ngoring Lake water body monitoring are Landsat series and Sentinel-2 satellite data. Landsat series satellites mainly include Landsat-5 TM, Landsat-7 ETM+, and Landsat-8 OLI, and their revisit periods are 16 days. The Landsat-5 [31] payload is a thematic mapper (TM) covering seven spectral bands. Except for the thermal infrared band (Band 6: 10.40–12.5 μm), which has a resolution of 120 m, the other bands have a resolution of 30 m. The sensor carried by the Landsat-7 satellite is the Enhanced Thematic Mapper (ETM+), with eight spectral bands, covering different wavelength ranges from the infrared to visible light. Compared with Landsat-5 TM, ETM + adds the Band 8, with a spatial resolution of 15 m. In addition, the Band 6 resolution is improved from 120 m to 60 m, and the overall accuracy is higher [31]. The sensors on Landsat-8 satellite are the Operational Land Imager (OLI) and Thermal Infrared Sensor (TIRS) [32]. OLI has nine spectral bands: the resolution of panchromatic band (Band 8: 0.500–0.680 μm) is 15 m, and that of the other bands is 30 m. Compared with Landsat-7

ETM+, OLI increases the coastal band (Band 1: 0.433–0.453 μm) and Cirrus band (Band 9:0.136–1.390 μm). Band 1 is mainly used for coastal area observation, and Band 9 is used for cirrus cloud detection [31].

Sentinel-2 is a high-resolution multispectral imaging satellite and divided into two satellites (2A and 2B) with a revisit period of five days. Its payload is the multispectral imager (MSI), which covers 13 spectral bands with a width of 290 km. The spatial resolutions of different bands are 10 m, 20 m, and 60 m, respectively, of which the spatial resolutions of blue (Band 2), green (band 3), red (Band 4) and near-infrared (Band 8) are 10 m. Sentinel-2 is mainly used for the observation of forest vegetation, inland water bodies, land cover, coastal, and offshore waters [33]. In order to ensure the accuracy of the experimental results, remote sensing images with a cloud cover of less than 5% were used in this study, with a total of 135 scenes. The specific data sources are shown in Table 2.

**Table 2.** Summary of Landsat Series and Sentinel-2 Image Data in Ngoring Lake.

| Satellite Name | Observation Time | Track Number | Quantity |
| --- | --- | --- | --- |
| Landsat-5 TM | 1992–2011 | 134_36 | 60 |
| Landsat-7 ETM+ | 1999–2002 | 134_36 | 17 |
| Landsat-8 OLI | 2013–2021 | 134_36 | 38 |
| Sentinel-2 A | 2021.01–2021.11 | N0301_R090 | 8 |
| Sentinel-2 B | 2021.01–2021.11 | N0301_R090 | 12 |

2.2.3. Satellite Altimeter Data

The information on the lake water level mainly comes from Hydroweb (http://Hydroweb.theia-land.fr, accessed on 1 March 2022). Hydroweb is a data center jointly established by the LEGOS Laboratory in Toulouse, France, and the HYDROLARE project under the responsibility of the National Institute of Hydrology (SHI) of the Russian Academy of Sciences [34]. The data center can provide water level timing information of approximately 150 lakes and reservoirs around the world free of charge. The historical water level data obtained are based on and fused with data from multiple satellite sensors, including TOPEX/Poseidon (1992–2005); ERS-1 (1991–1996); ERS-2 (1995–); GFO (2000–); Jason-1 (2001–2013); Jason-2 (2008–); Jason-3 (2016–); Envisat (2002–); Saral (2002–); and Sentinel-3A (2016–) [35].

*2.3. Methods*

2.3.1. ICESat-2 Water Level Extraction and Multisource Altimeter Data Fusion

All data products of ICESat-2 are free to the public in HDF5 format [29]. The longitude and latitude coordinate information of the corresponding photon points and other parameters as well as "ht_water_surf, segment_geoid, water_depth" are extracted from the "/gtxx" data group in the ATL13 data product. At the same time, the track points of ICESat-2 on the corresponding date are extracted. The along-track footprint points are transformed into ShapeFile files using ArcGIS. ICESat-2 observation data of all those crossing Ngoring Lake basin from September 2018 to June 2021 are obtained. The data of ICESat-2 are represented as discrete photon point clouds, and the laser beam transmitted to the ground has a certain deviation. The spot points may fall on the land at the boundary of the lake, and the obtained data have errors. Therefore, for each group of extracted data values, the elevation data that differ from most of the water level values by more than 20 cm are excluded [36].

In order to verify the height measurement accuracy of an altimeter, it is necessary to unify the reference datum before the fusion of the water level time series [37]. The water level values retrieved by different spaceborne altimeters have a certain degree of systematic deviation due to the differences in their own instruments and equipment and the various parameters of the operating orbit. The previous research method used the converted altimeter water level, selected the water level on the same date, established the correlation equation through linear fitting, and constructed the long-term water level series of a single lake [38]. The reference ellipsoid of the water level value was extracted

in ICESat-2 is WGS84 and EGM96 leveling surface [39], whereas the reference ellipsoid of LEGOS HYDROWEB is the GRACE Gravity Model 02 (GGM02) [34]. Subsequently, through the average difference value, the fused altimeter water level value is used to construct the Ngoring Lake water level sequence of the multisource spaceborne altimeter, and the correlation between the lake water level and water storage capacity is studied.

In addition, to ensure the accuracy of the research results, the differences between the observed water level values on the coincidence date of ICESat-2 and Hydroweb are compared. Three statistics of linear regression analysis ($R^2$), mean absolute error (*MAE*), and root mean square error (*RMSE*) are used to verify and analyze the accuracy. The specific formula is as follows:

$$R^2 = 1 - \frac{\sum_{i=1}^{n} (y_i - y_i')^2}{\sum_{i=1}^{n} (\bar{y}_i - \bar{y}_i)^2} \tag{1}$$

$$MAE = \frac{1}{n} \sum_{i=1}^{n} |y_i - y_i'| \tag{2}$$

$$RMSE = \sqrt{\frac{1}{n} \sum_{i=1}^{n} (y_i - y_i')^2} \tag{3}$$

In Formulas (1)–(3), $y_i$ is the water level observed by ICESat-2, $y_i'$ is the observed water level on the same date corresponding to Hydroweb and ICESat-2, $\bar{y}_i$ and $\bar{y}_i$ are the average values of $y_i$ and $y_i'$, respectively, $i$ is the index of the observed values, and $n$ is the number of overlapping days of water level observations.

2.3.2. Accuracy Verification and Evaluation of ICESat-2 and Satellite Altimeter

Different altimeters have systematic differences in orbit height, orbit inclination, and return period. There are also some differences between the water level measurement results of different altimeters. Therefore, for lakes covered by two or three water-level sources, the water level observations should be converted to a uniform baseline value to form a denser observation sequence. The correlation between the results of different types of water level is constructed to achieve this goal. The correlation is established according to the results obtained in the overlapping period. Only when the overlapping period has more than 10 pairs of data and the related $R^2$ value is greater than 0.88, the correlation between the observation results of the two altimeters is considered to be credible. If the $R^2$ value is less than 0.88, then the level results are selected with a long-time span for the next analysis [40].

Figure 4a shows the change in water level corresponding to the coincidence date of the ICESat-2 water level value and the satellite altimeter from October 2018 to June 2021. The figure shows that the change trend of the ICESat-2 altimetry water level is basically consistent with that observed by satellite altimeter. There are 19 overlapping observation days between the ICESat-2/ATL13 observation data and Hydroweb data used in this study. Figure 4b shows the absolute elevation difference after the unified coordinate system of ATL13 data and Hydroweb provided water level value. The maximum absolute elevation difference value is 0.21 m (22 March 2019), the minimum absolute elevation difference is only 0.0008 m (15 December 2018), and the average absolute elevation difference is 0.049 m. Furthermore, the correlation coefficient index is used to evaluate the height measurement accuracy of the ICESat-2 lake water level. Figure 4c shows the relationship between the multisource satellite altimeter in Ngoring Lake and the lake water level values extracted by ICESat-2. The correlation between the two is very high, with an $R^2$ above 0.97, satisfying the prerequisite for $R^2$ values greater than 0.88; *MAE* = 0.42 m and *RMSE* = 0.077 m. The results show that ICESat-2 can be integrated with other satellite altimeters as an effective source of supplementary data to achieve a more intensive observation and research results.

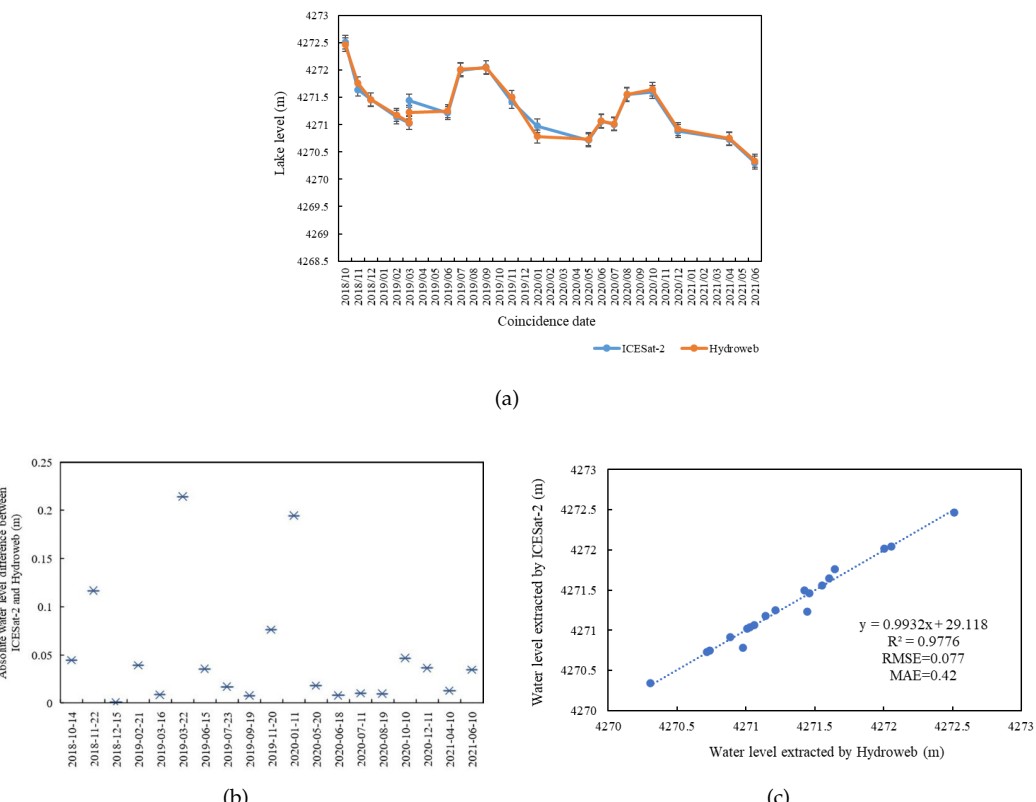

**Figure 4.** Comparative analysis of water level values extracted by ICESat-2 and Hydroweb. (**a**) Coincident date water level of ICESat-2 and Hydroweb; (**b**) Absolute difference of water level values of ICESat-2 and Hydroweb; (**c**) Correlation of water level values extracted by ICESat-2 and Hydroweb.

### 2.3.3. Extraction of Lake Surface Area

The normalized difference water index (*NDWI*) is used to extract the water bodies and obtain the water segmentation objects. *NDWI* is based on the spectral reflection intensity characteristics of vegetation and water in the visible and near-infrared bands and is constructed using the ratio of the reflectance of water in the green and near-infrared bands [41] to enhance the water information and suppress the vegetation information to a certain extent. The index can also effectively distinguish information, such as water bodies, vegetation, and mountain shadows [42]. The calculation formula is as follows:

$$NDWI = (Green - NIR)/(Green + NIR) \qquad (4)$$

In Formula (4): green is green band reflectivity, and *NIR* is near-infrared band reflectivity.

Ngoring Lake is an oligotrophic plateau lake with a clear water body [43], which satisfies the Class I water quality standard of "Surface Water Environmental Quality Standard" [39]. *NDWI* can achieve a better effect of water boundary extraction. Based on the preprocessed remote sensing images, the BandMath of the ENVI+IDL program is used to input the *NDWI* calculation formula, and the water body information is highlighted through the band ratio relation, as shown in Figure 5. The raster vector transformation of the segmented object is carried out using the Rastertopolygon tool in the Arcmap software to obtain the information of the berthing boundary of Ngoring Lake. Combined with manual visual interpretation, the final surface area of the lake is calculated.

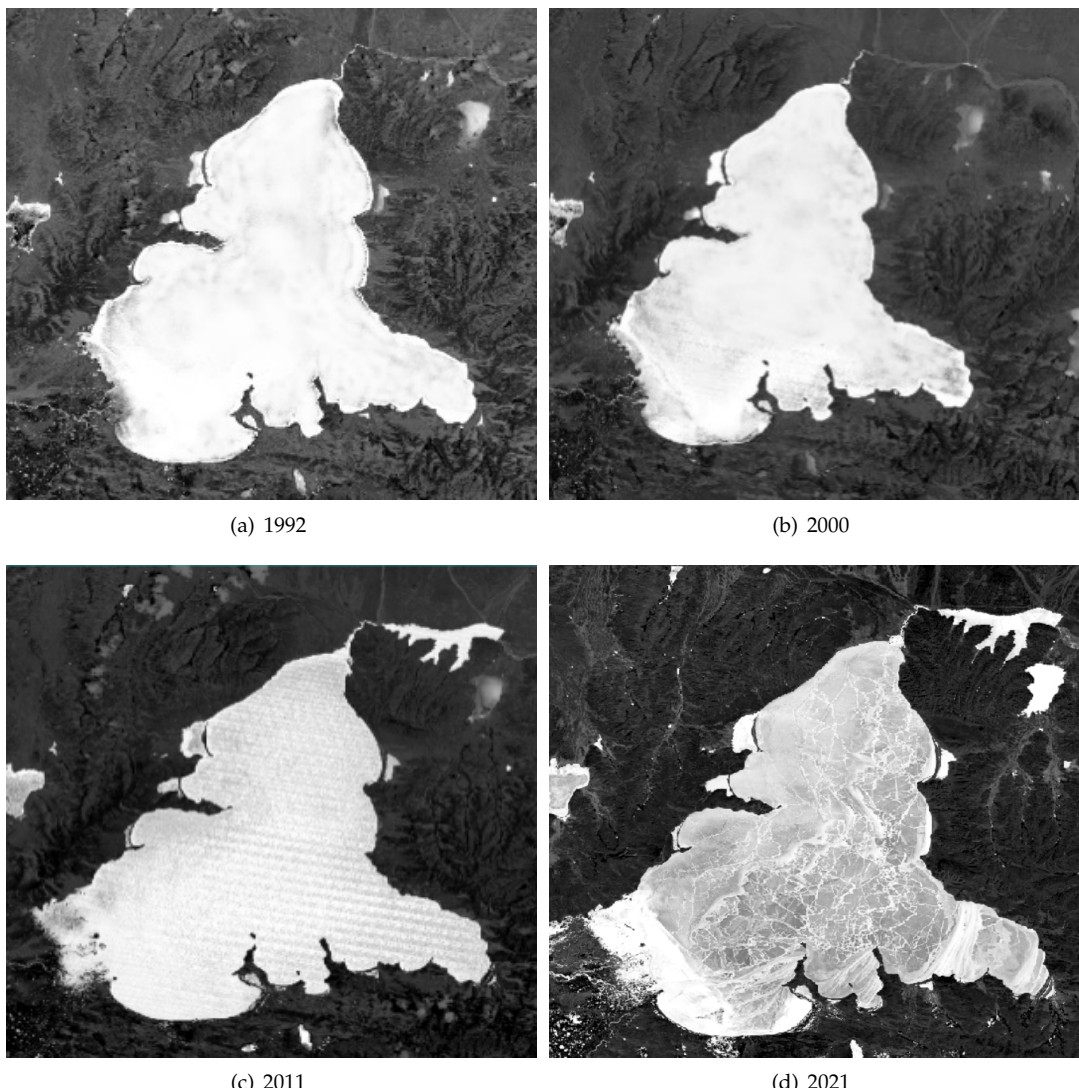

(a) 1992

(b) 2000

(c) 2011

(d) 2021

**Figure 5.** Water body extraction by *NDWI*.

2.3.4. Estimation of the Lake Water Storage Based on Water Level—Area Relationship

The water storage capacity of the lake cannot be obtained by a direct measurement. It needs to be indirectly estimated by constructing the lake water level–water storage relationship model with the help of remote sensing monitoring or the measured lake area, water level, and water depth (underwater topography) [44].

The main factors affecting the net budget of the lake water include the dynamic changes in the lake surface area and water level. When the water level rises, the area of the lake increases. On the contrary, if the water level falls, the lake area shrinks [45]. Therefore, the change in water storage capacity can be estimated by the correlation between the water level and lake area. On this basis, the following calculation formula proposed by Taube is adopted [46]:

$$\Delta V = \left( \frac{1}{3} L_{i+1} - L_i \right) \times \left( \sqrt{S_{i+1} S_i} + S_{i+1} + S_i \right) \tag{5}$$

In Formula (5), $\Delta V$ represents the change value of the lake water storage capacity when the lake water level changes from $L_i$ to $L_{i+1}$, that is, the net income and expenditure of the lake: $S_i$ and $S_{i+1}$ are the surface areas of lakes in the $i$th stage and $i + 1$th stage, respectively. By accumulating the net income and expenditure of the lake step by step, the water balance data of Ngoring Lake from 1992 to 2021 can be obtained.

## 3. Results

### 3.1. Analysis of the Changes in Water Level and Water Storage Capacity in Ngoring Lake

The average altitude of Ngoring Lake basin is above 4200 m, which belongs to alpine steppe climate [47]. Winter is long, cold, dry, and windy, while the summer is short, cool, and rainy, which has different response than alpine meadows [48]. Figure 6 shows the changes in water level and water storage capacity of Ngoring Lake from 1992 to 2021. Evidently, the fluctuation of the water bodies of Ngoring Lake during the past three decades is relatively complicated, and significant differences are found in the change rate of water level and water storage capacity during different periods.

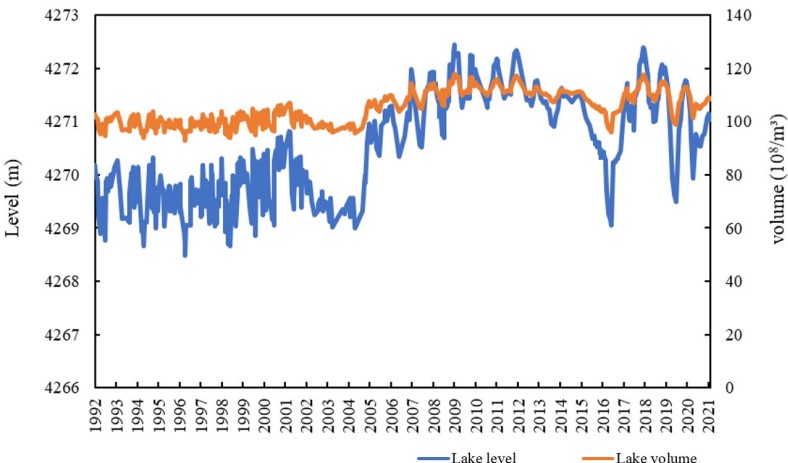

**Figure 6.** Overall variation trend of water level in Ngoring Lake from 1992 to 2021.

Figure 7 shows the fluctuation of the water level and water storage capacity of Ngoring Lake in different years. The specific analysis is as follows:

During the period, 1992–2001 (Figure 7a), the annual variation of the water body was relatively stable. The initial water level on 12 October 1992 was 4270.19 m, and the observed water level on October 23, 2001 was 4270.51 m, which fell to the lowest water level in January 1997, at approximately 4268.49 m. The water level of the lake rises at a change rate of 0.041 m/y, and the water storage capacity increases slowly at 0.023 TMC/y.

From January 2002 to January 2005 (Figure 7b), the water body of Ngoring Lake showed a sharp decreasing trend, the water level decreased rapidly by −0.451 m/y, the water storage capacity decreased by −2.753 TMC/y, and the average annual water level was approximately 4269.405 m.

From May 2005 to May 2015 (Figure 7c), the water level of Ngoring Lake rose rapidly at the end of May 2005 and reached the highest water level in October 2009, which was approximately 4272.44 m. Then, the growth rate slowed down. The overall water level growth rate was approximately 0.207 m/y, and the water storage capacity growth rate was approximately 1.277 TMC/y. During this period, it was mainly affected by the hydropower station located in the lower reaches of Ngoring Lake. The Yellow River Source Hydropower Station, which was completed in 2001 and passed the project completion acceptance on 21 July 2006, controlled the runoff out of the lake, reduced the water loss of the lake, and led to the rapid rise of the water level of Ngoring Lake [24].

From August 2015 to August 2017 (Figure 7d), the water level of Ngoring Lake dropped sharply, and reached the lowest water level of 4269.05 m on 18 March 2017. The change rate of the water level was −0.545 m/y and the change rate of water storage was −3.369 TMC/y. This is mainly due to the construction of a reservoir to store water in Zaling Lake, which is in the same section of the Yellow River, resulting in a rapid drop in the water level of Ngoring Lake downstream [45].

From September 2017 to November 2021 (Figure 7e), the water level of Ngoring Lake shows a fluctuating growth state as a whole, and the trend of a rising water level signifi-

cantly slows down. The cumulative increase was approximately 0.61 m, the water level change rate was 0.045 m/y, and the water storage volume change rate was 0.288 TMC/y.

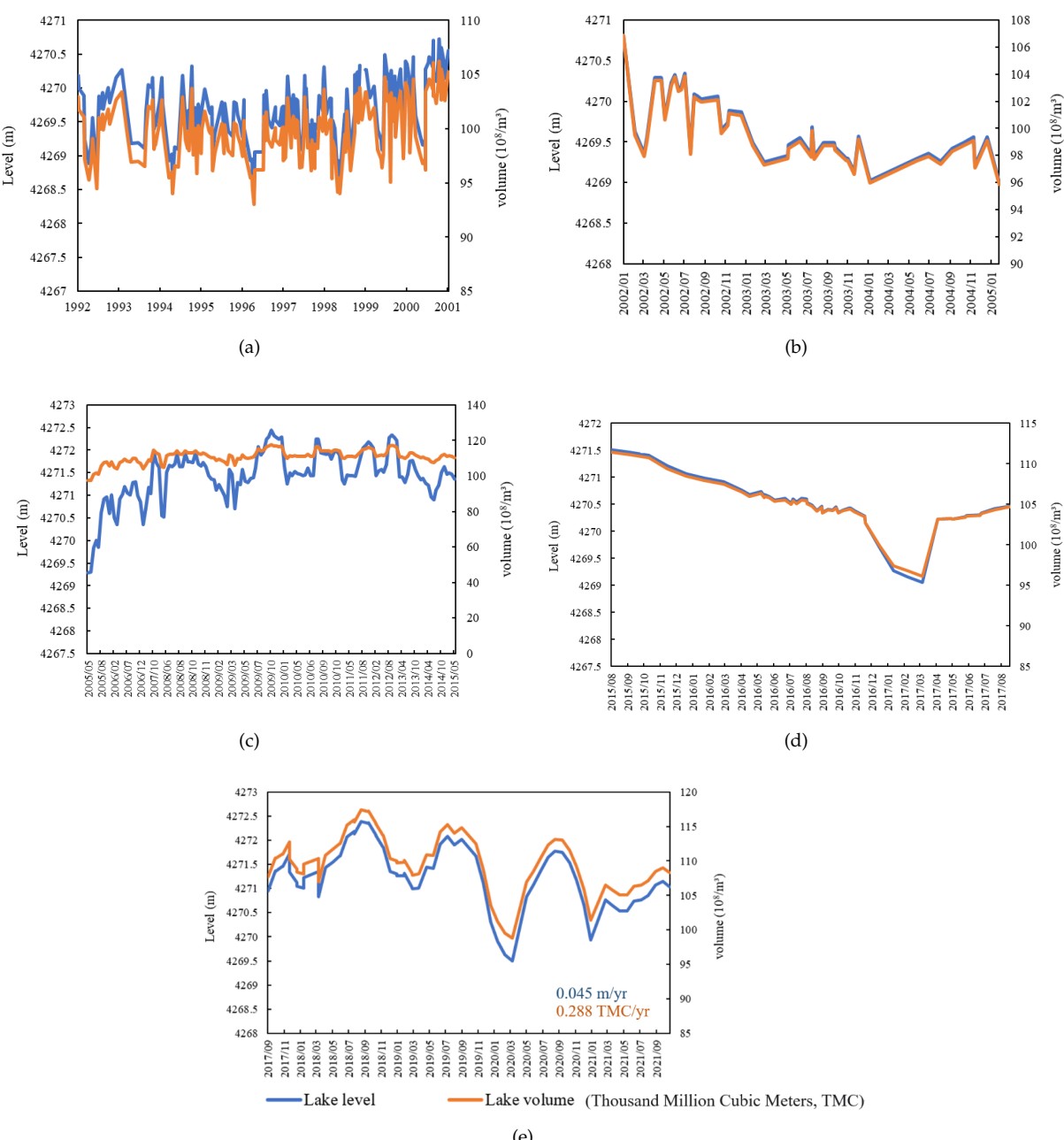

**Figure 7.** Fluctuation of the water level (blue lines) and water storage capacity (orange lines) of Ngoring Lake in different years. (**a**) Changes in the water level and volume of Ngoring Lake from 1992 to 2001; (**b**) Changes in the water level and volume of Ngoring Lake from 2002 to 2005; (**c**) Changes in the water level and volume of Ngoring Lake from 2005 to 2015; (**d**) Changes in the water level and volume of Ngoring Lake from 2015 to 2017; (**e**) Changes in the water level and volume of Ngoring Lake from 2017 to 2021.

A peak period of the water level change is observed in Ngoring Lake from April to May due to the increase in temperature in spring, the confluence of surface runoff caused by alpine ice and snowmelt water. The peak period of the annual water level caused by the increase in summer precipitation is generally from August to October [47].

January, May, and September are selected as the representative months of the water level of the lake during the freezing period, melting period, and nonfreezing period, respectively, to further analyze the change in water level in Ngoring Lake during the year.

Figure 8 shows that, in most years, the water level during the nonfreezing period was higher than that in the freezing period. Only in 1998, 2000, 2003, 2010, and 2016, the average water level value in January was higher than that in September, and the maximum absolute difference is approximately 0.527 m. During the period from 1995 to 2016, the water level in May was higher than in January and September, and then gradually decreased. Only in 2016 was the monthly average water level higher than in September, and even the average monthly water level value during the period 2010–2016 was lower than in January. Combined with the linear regression analysis, in general, the order of the monthly average water level change rate from large to small is as follows: non-freezing period, melting period, and the freezing period (blue–green–yellow).

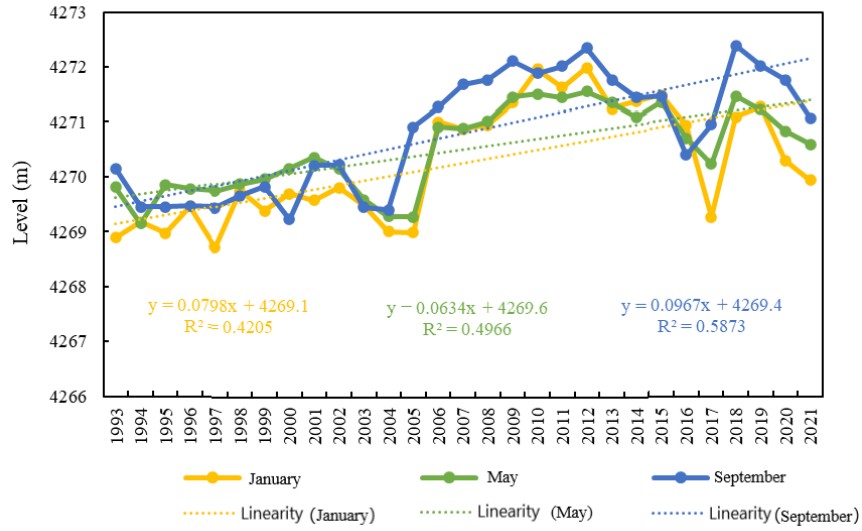

**Figure 8.** The monthly average water level of Ngoring Lake during glacial, melting, and nonglacial periods during the period 1993–2021.

*3.2. Changes in Surface Area and Lake Boundary of Ngoring Lake*

From 1992 to 2021, the overall lake area of Ngoring Lake showed a fluctuating upward trend (Figure 9). The lake area was 625.35 km² in October 1992 and 643.7 km² in November 2021, representing an increase of approximately 0.633 km²/y. In January 1997, the lake area was at its minimum, approximately 590.25 km², and in October 2009, the lake surface area expanded to the historical maximum, at approximately 675.15 km².

From 1992 to 2001, the overall surface area of Ngoring Lake showed a slow growth trend. From 2002 to 2005, the lake area decreased rapidly, and the average change rate was approximately −10.67 km²/y. From June 2005 to June 2015, the area of Ngoring Lake rapidly increased and reached the maximum lake area during this period in October 2009, at approximately 675.15 km².

During the period from May 2016 to May 2017, the surface area of Ngoring Lake decreased sharply. In February 2017, the surface area of the lake decreased to 603.82 km² and gradually recovered in June of the same year. In March 2020, it dropped to 610.84 km² again, and then the surface area showed a slow growth trend.

According to the shoreline change of Ngoring Lake during the past 30 years (Figure 10a), the change in lake surface area mainly occurs in the southwest corner and northeast, which are the estuary of the source of the Yellow River and the artificial interception reservoir area, respectively, and the changes in the lake area are more sensitive than in other areas. Figure 10b shows the change in the shoreline of the Ngoring Lake basin from

1992 to 2001, which gradually expanded outward in the southwest corner. Figure 10c shows the change in shoreline of Ngoring Lake from 2001 to 2005. Due to the artificial construction of the reservoir, the runoff converges in the northeast to form a water area connected with the main body of Ngoring Lake, and the significant change area of the lake is still located in the southwest. Figure 10d shows the shoreline changes during the period of maximum lake surface area (2009) and minimum lake surface area (1997) of Ngoring Lake from 1992 to 2021, and the change area is consistent with the overall shoreline change area.

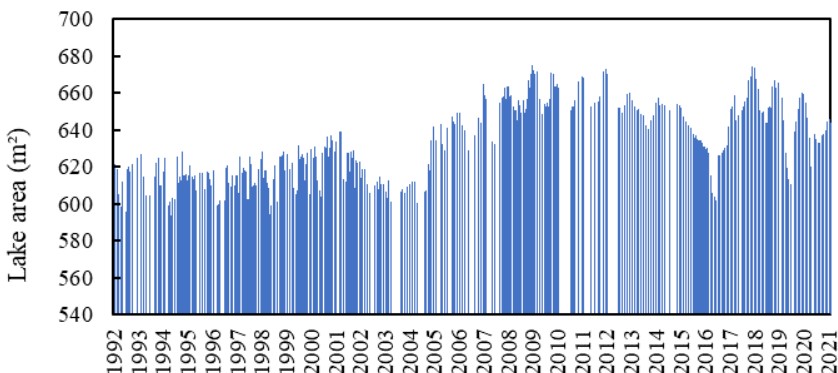

**Figure 9.** Surface area change trend of Ngoring Lake during the period 1992–2021.

The change in lake area is mainly affected by the natural environment and human activities [20]. The water surface change in Ngoring Lake in the last 30 years is the result of the comprehensive action of many factors, such as climate change, human activities, and geological tectonics [49]. The natural environment is mainly dominated by rainfall and temperature and climate factors. According to the analysis of observation data from Maduo Meteorological Station, the temperature of Ngoring Lake has gradually increased over the past three decades, and the temperature has accelerated since the 1990s. The precipitation decreased in the 1990s and slightly rebounded after 2000, and there was no obvious change trend in precipitation as a whole. The relative humidity has been decreasing. This finding shows that the regional climate of Ngoring Lake has become warm and dry in the past 30 years [50], thereby directly affecting the change in surface area of Ngoring Lake.

The Ngoring Lake basin is located in an alpine climate zone, with a large proportion of no man's land. During the period of 1992–1998, the influence of human factors was small, and the influence of the natural environment became the main reason for the change in the lake water level in this area. After 2000, the surface area of Ngoring Lake expanded again, which was mainly caused by the construction of a hydropower station at the outlet of Ngoring Lake and the improvement of the lake water level, in which human activities played a leading role [47].

*3.3. Correlation and Attribution Analysis between the Surface Area, Water Level, and Water Storage Changes in Ngoring Lake*

3.3.1. Correlation among Surface Area, Water Level, and Water Storage Capacity

Figure 11a–c shows that the water level, the lake surface area, and the water storage capacity have a statistically significant correlation, respectively. Among them, the correlation $R^2$ among the water level value and lake surface area, lake surface area, and water storage capacity is as high as 0.9998, and the correlation $R^2$ between water level value and lake water storage capacity is as high as 0.9992. Therefore, reconstructing the water level value or surface area is highly accurate at the corresponding time based on the water level value or lake surface area to estimate the lake water storage change information.

Figure 12 shows the general trend of water level, water area and water storage capacity of Ngoring Lake from 1992 to 2021.

### 3.3.2. Analysis of the Influence of Meteorological Factors on Water Level

Figure 13 shows the monthly data of precipitation and runoff in the source region of the Yellow River from 2003 to 2015. The annual average precipitation in the source region of the Yellow River is 562 mm, and the interannual variation of precipitation is insignificant, but it exhibits evident seasonal changes. According to the data of Tangnaihai Hydrological Station, the annual average runoff depth in the source area of the Yellow River from 2003 to 2015 was approximately 161 mm. In 2012, the precipitation in the source region of the Yellow River reached 630 mm, which was higher than the average precipitation of 562 mm from 2003 to 2015. In the same year, the runoff of the Yellow River Basin increased to the maximum. This finding is consistent with the phenomenon that the water level of Ngoring Lake increased in 2012 (Figure 14).

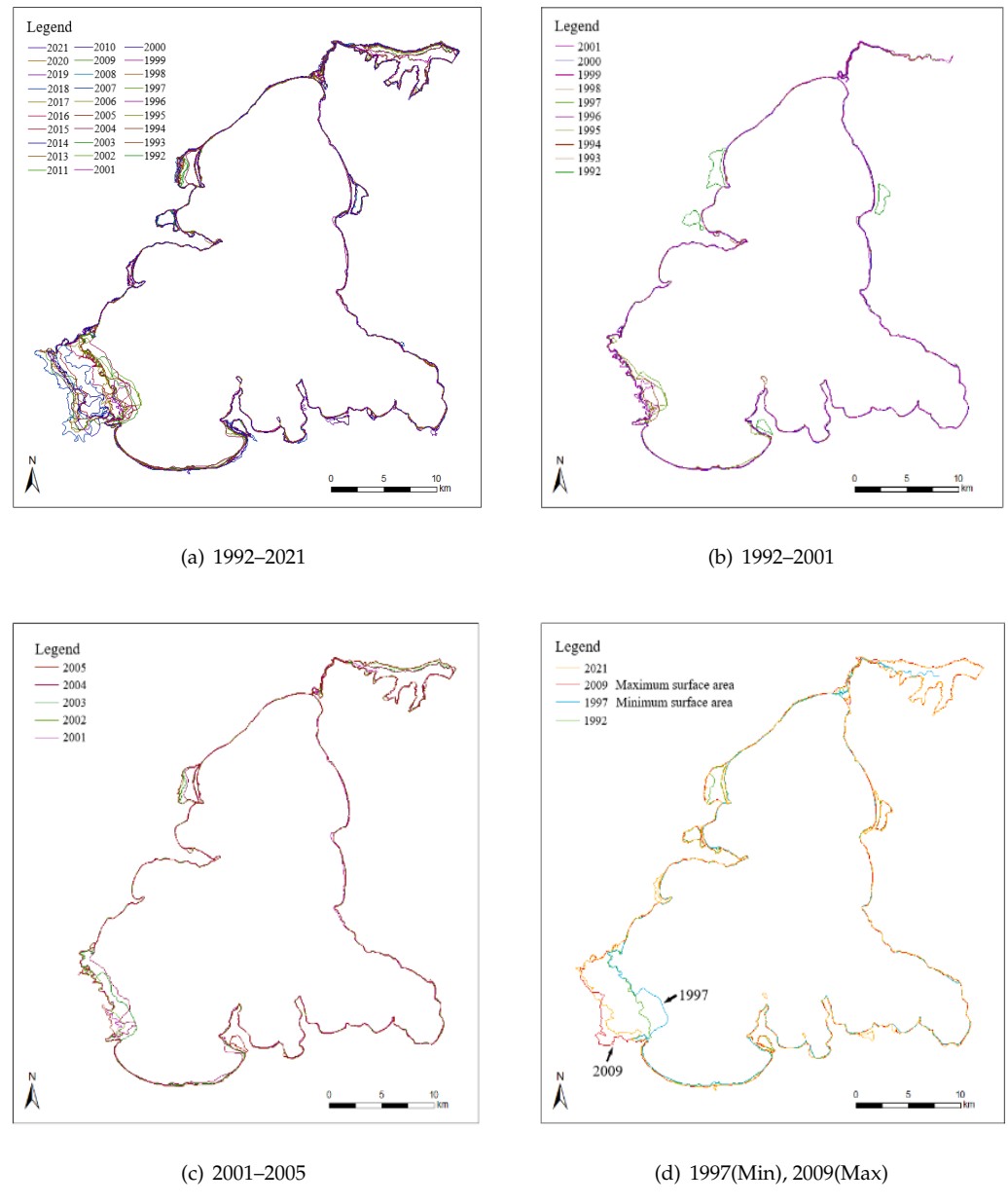

(a) 1992–2021

(b) 1992–2001

(c) 2001–2005

(d) 1997(Min), 2009(Max)

**Figure 10.** Shoreline changes of Ngoring Lake during the period 1992–2021.

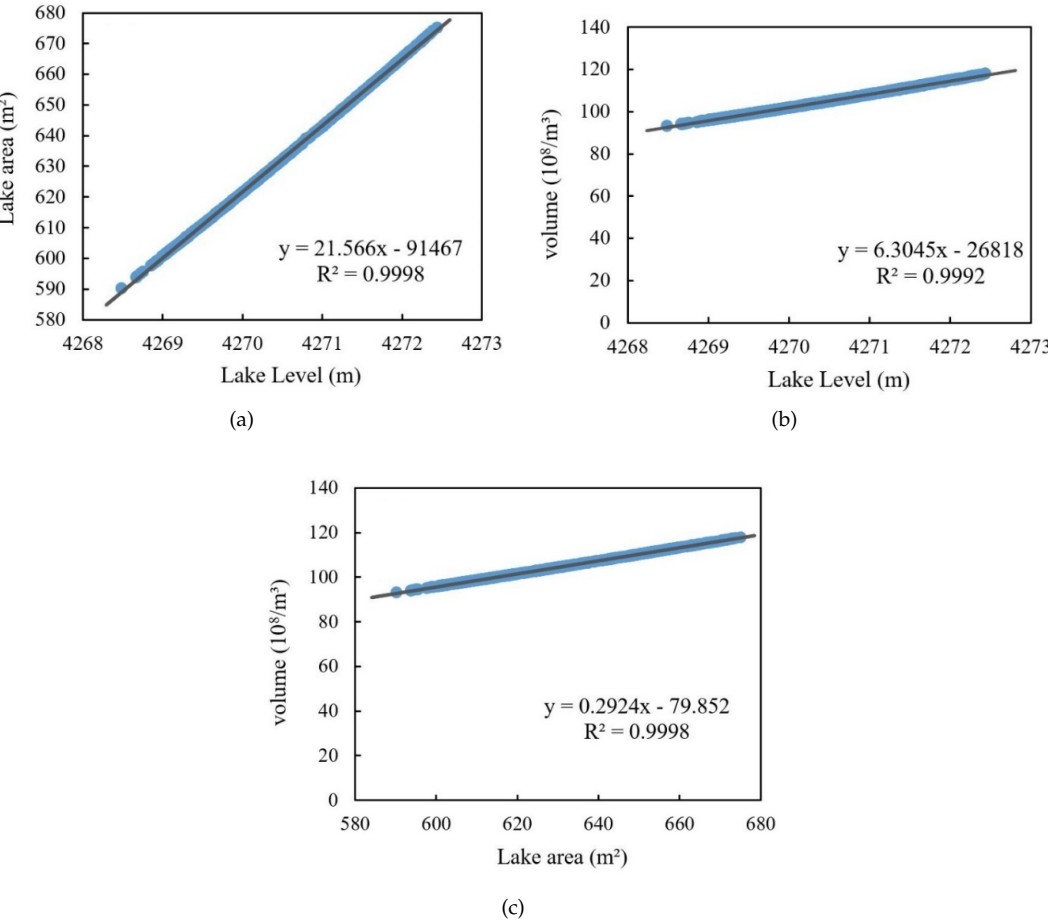

(a)

(b)

(c)

**Figure 11.** Correlation evaluation of the surface area, water level, and water storage capacity of Ngoring Lake.

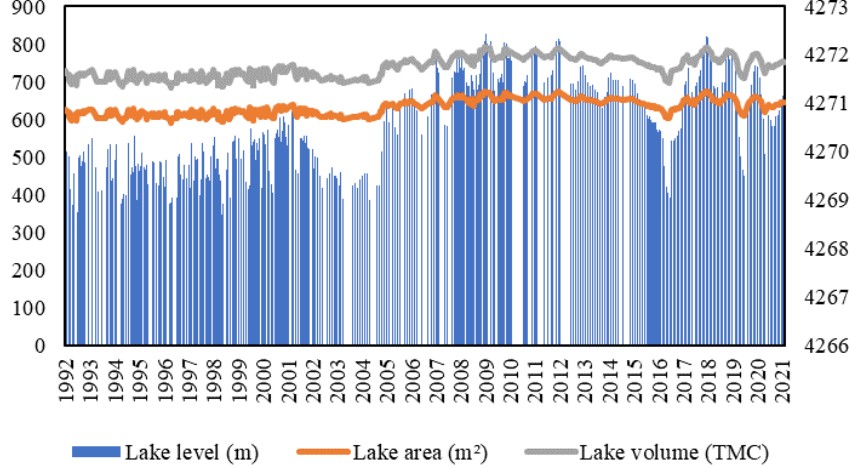

**Figure 12.** Observation results of water level, surface area, and water storage capacity of Ngoring Lake during the period 1992–2021.

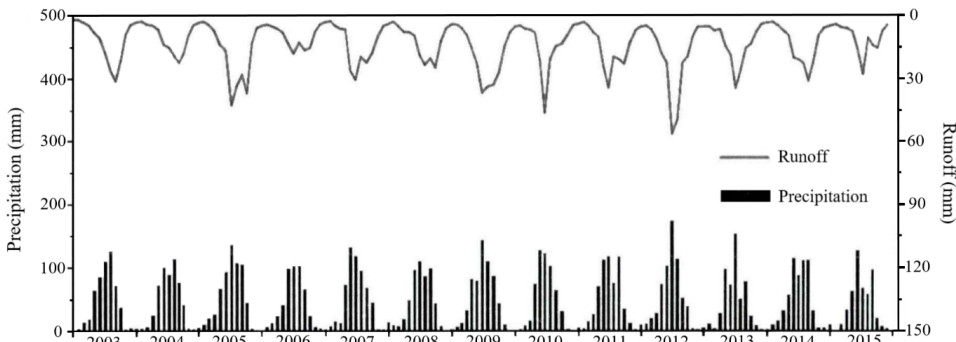

**Figure 13.** Monthly time series of precipitation and runoff across the Source Region of the Yellow River during the period 2003–2015 [47].

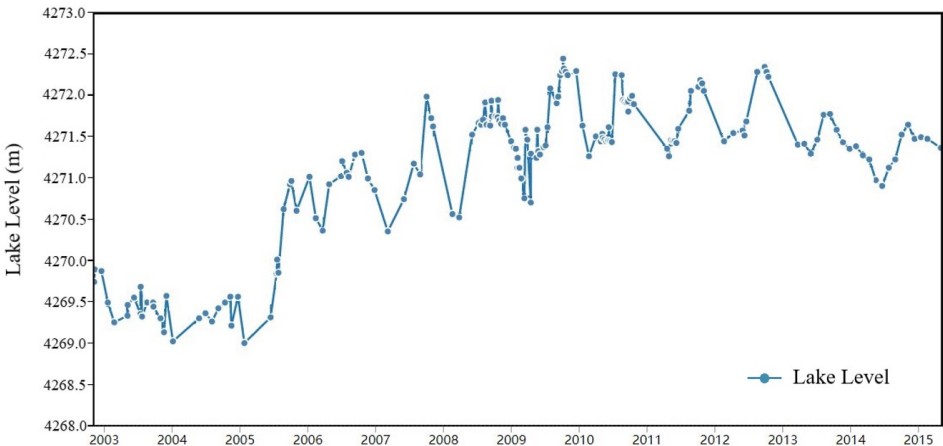

**Figure 14.** Monthly average water level of Ngoring Lake during the period 2003–2015.

As shown in Figure 15a, during the period 2003–2015, the average annual water storage in the source region of the Yellow River had an evident upward trend, and the average annual water level increased at a rate of 1.97 mm/y. Figure 15b shows that the annual precipitation in this region has remained stable and has not increased significantly.

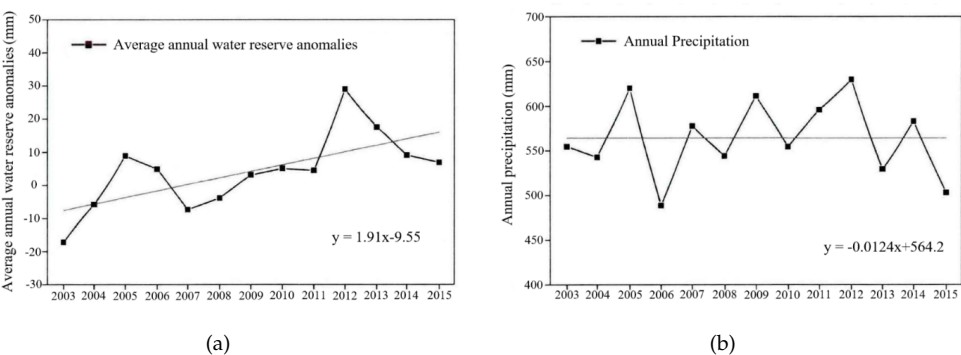

(a)

(b)

**Figure 15.** Annual time series of (**a**) terrestrial water storage anomalies; and (**b**) precipitation during the period 2003–2015 across the entire Source Region of the Yellow River [47].

Figure 16a shows the annual average temperature of Maduo Station at the source of the Yellow River from 1990 to 2017. The overall temperature change showed an evident warming trend, showing the characteristics of a climate alternating between cold and warm, fluctuating and rising, and the change rate was 0.31 °C/10a (annual). Figure 16b shows the annual average temperature anomaly at Maduo Station. In 1998, the temperature changed

from a negative anomaly to a positive anomaly and reached the highest value in history in 2016.

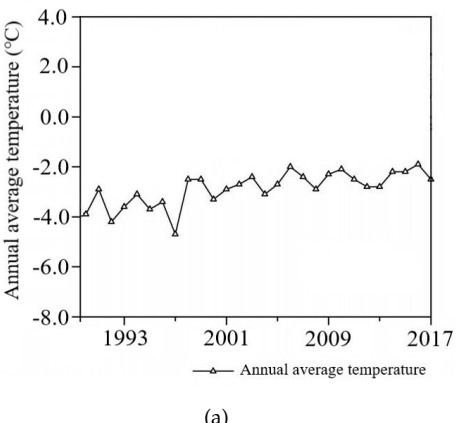

(a)

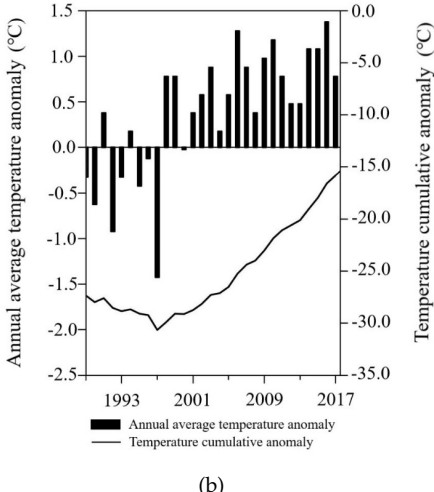

(b)

**Figure 16.** (**a**) Annual mean temperature at Maduo station in the source region of the Yellow River; and (**b**) annual average temperature anomalies at Maduo station during the period 1990–2017 [51].

Table 3 shows the comparison between the climatic factors of the source of the Yellow River and the corresponding annual average water level change in Ngoring Lake during the period of 2003–2013. The results show that climate change is still an important factor affecting the change in water reserves in most sub-basins. At the same time, human activities have an evident effect on the change in water reserves in the study basin, especially in the middle and lower reaches of the Yellow River [52]. The establishment of reservoirs and the implementation of other water conservancy projects have led to great changes in the annual water reserves in some basins, and then affect the changes of the water body of a single lake. In 2006, the Yellow River Source Hydropower Station was officially put into operation, and the annual average water level of Ngoring Lake increased from 4269.62 m in 2005 to 4270.94 m. Since then, from 2007 to 2013, the average annual water level of Ngoring Lake has remained above 4271.00 m, whereas the annual average temperature and annual precipitation have not significantly changed. The finding shows that compared with the effect of climate change on the change in water storage in the basin, human activities, such as water conservancy project construction, have a more significant effect on the change in lake water storage.

**Table 3.** Climatic factors at the source of the Yellow River and changes in the average annual water level of Ngoring Lake during the period 2003–2013.

| Year | Annual Average Water Reserve Anomaly | Annual Average Temperature | Annual Precipitation | Annual Average Water Level |
|---|---|---|---|---|
| 2003 | −18.4 mm | −2.51 °C | 552 mm | 4269.39 m |
| 2004 | −6.1 mm | −3.12 °C | 541 mm | 4269.34 m |
| 2005 | 9.2 mm | −2.94 °C | 628 mm | 4269.62 m |
| 2006 | 5.6 mm | −2.14 °C | 483 mm | 4270.94 m |
| 2007 | −7.5 mm | −2.53 °C | 578 mm | 4271.12 m |
| 2008 | −4.8 mm | −3.06 °C | 542 mm | 4271.40 m |
| 2009 | 4.6 mm | −2.41 °C | 619 mm | 4271.68 m |
| 2010 | 5.2 mm | −2.21 °C | 551 mm | 4271.69 m |
| 2011 | 4.9 mm | −2.62 °C | 596 mm | 4271.72 m |
| 2012 | 29.6 mm | −2.93 °C | 632 mm | 4271.84 m |
| 2013 | 18.2 mm | −2.91 °C | 530 mm | 4271.49 m |

## 4. Discussion

### 4.1. Water Level–Water Storage Inversion Error Analysis

As a large freshwater lake on the TP and the upper reaches of the Yellow River, Ngoring Lake not only regulates the local climate through the "lake effect", but also directly affects the seasonal variation of the waters of the lower Yellow River. This study makes full use of the continuity of medium- and high-resolution Landsat series images, and combines them with higher resolution Sentinel-2 images to provide more accurate and long-term monitoring of the water body of Ngoring Lake. Meanwhile, the water level data obtained by satellite altimetry were used to convert the changes in lake area based on Landsat series images into changes in water storage.

For Landsat series and Sentinel-2 images, most of the water bodies in the study area can be successfully and effectively extracted using the MNDWI and *NDWI* water body index proposed by Xu [32,41]. MNDWI is more suitable for extracting the water body boundaries of eutrophic lakes; for shallow water depths, MNDWI also has certain errors; for oligotrophic Ngoring Lake, the effect of extracting lake boundaries by *NDWI* and MNDWI index is not significant. In addition, the water body boundaries extracted using the optimized threshold are more reasonable than those using a uniform threshold of 0, which effectively avoids the phenomenon of mistakes and omissions in some areas. Due to Sentinel-2 images having higher spatial resolution, the overall accuracy of the water body extraction is more precise. It is more accurate for shallower water extraction, and the overall extraction effect is better than the Landsat series of images. However, it is also more susceptible to other factors such as the weather conditions, water quality, and phytoplankton on the water surface, which may cause the misjudgment of the water body. Therefore, the appropriate water body index should be carefully selected according to the water body condition in the study area to ensure the best water body extraction effect.

The main uncertainty of the estimation of lake water storage capacity in this study lies in the failure to obtain the specific measured water depth data of Ngoring Lake and the fusion of water level data from different satellite altimeters. In this study, the water level data provided by HYDROWEB were consulted to validate the ICESat-2 based water level estimation. ICESat-2 currently has a small amount of data and a long time interval. There is a certain time interval between the data measurement of ICESat-2 and HYDROWEB, and the height variation between these time periods may cause the uncertainty of data comparison. The results show that the lake level trends of the two datasets are highly correlated, with an $R^2$ value of 0.9776 (*RMSE* = 0.077). The water level data of Ngoring Lake extracted by ICESat-2 are slightly higher than that of the HYDROWEB dataset.

The results of the study show that the water level, lake area and water storage of Ngoring Lake show a fluctuating increasing pattern from 1992 to 2021. The water level and

area reached the lowest in 2017. Since then, Ngoring Lake has slowly started to expand. Overall, the present study is more consistent with the findings of Luo et al. [53]. However, due to the lack of lake bathymetry datasets, the water storage estimated in this study has a certain error with the real water variation. In addition, different altimetric data have uncertainty due to their different data quality. In the future, we will consider combining the actual bathymetric data of the lake and fusing different altimetric satellite data to extend the research on water level and storage capacity.

### 4.2. Possible Driving Forces of Lake Changes

Changes in lake level and water volume are the result of a combination of climate variability and anthropogenic factors. Existing studies have proven that in recent years, the internal climate change in the TP has led to the gradual increase in precipitation, and the rise of temperature has led to the increase in glaciers and permafrost melt water, which together contributed to the rapid expansion of lakes in recent decades [53–56]. The melting water of glaciers and snow became an important water supply for plateau lakes [55].

According to the research results, the pattern of the lake water level rising or falling is basically consistent with the changes in temperature and precipitation of different degrees, which indicates that lakes in plateau areas are often more vulnerable and sensitive. In addition, the Ngoring Lake basin is more influenced by local economic construction and water conservancy projects, and the influence of human factors on Ngoring Lake is significantly greater than that of natural factors.

Considering the increasing contribution of glaciers and precipitation to the water balance, the water volume of inland lakes in the plateau is expected to continue to increase in the coming decades. This indicates that the water storage capacity of Ngoring Lake will also increase in the near future. The continuous rise in the water level and the expansion of the water area may breed a better ecological environment and richer biodiversity [56], which will be conducive to ecological protection and water and soil conservation in the Ngoring Lake basin.

### 5. Conclusions

In this study, combined with ICESat-2 satellite altimetry data, Hydroweb platform, and multisource optical remote sensing images, the long-time series water fluctuation monitoring of Ngoring Lake basin is realized. The long-term variation trend of the surface area of the Ngoring Lake basin can be obtained using the image data of the Landsat series and sentinel-2 from 1992 to 2021. The ICESat-2 observation data from 2018 to 2021 were integrated with the multisource satellite altimetry data provided by Hydroweb. More accurate observation results can be obtained by applying these to lake water level monitoring, which is of great significance to invert the water level observation data of non-measured areas and prolong the water-level observation time series of discrete water level stations. Based on the analysis of the water fluctuation in the Ngoring Lake basin in the past 30 years, the conclusions are as follows:

(1) The satellite images obtained from the Tibetan Plateau have high cloud coverage and a long return period. Thus, continuously monitoring the changes in lakes and water bodies in the whole plateau by only using a single remote sensing sensor is impossible. In this study, using the ICESat-2 photon counting laser altimeter data and multisource satellite altimeter combined with optical remote sensing satellite images, the dynamic data of the water level and area in Ngoring Lake basin were captured conveniently and accurately, and the changes in lake water storage were estimated. This method has high applicability to areas where the measured water level data cannot be obtained.

(2) In the last 30 years, the changes in water level, lake area, and water storage capacity of Ngoring Lake slightly increased, in general, from 4270.19 m, 625.35 km², and 102.92/billion m³ in 1992 to 4271.15 m, 646.1 km², and 109.01/billion m³ in 2021. The water storage capacity is slowly growing at approximately 0.21/billion m³ per year. During the research period from 1992 to 1998, the water fluctuation in Ngoring Lake was mainly af-

fected by natural environmental factors, such as the change in temperature, annual rainfall, and annual evaporation. After 1998, to satisfy the needs of local economic development, water conservancy projects, such as the construction of hydropower stations, have a more significant effect on the water body of Ngoring Lake.

(3) Lake water level monitoring is mostly based on spaceborne altimeters, such as the Jason series, Envisat, ICESat, and Cryosat-2. Most studies are mainly focused on lakes with an area of 100 km² or larger due to a long time span, irregular ground trajectory points, blank observation data, and other reasons, and less observation being made for smaller water bodies. According to the research results of reference [27], compared with the aforementioned satellite altimeter, ICESat-2 can realize the observation of lakes with a surface area of less than 0.1 km². In our study, ICESat-2/ATL13 data were analyzed and compared with the water level values obtained from Hydroweb. The water level values of coincidence dates were $R^2 = 0.9776$, $MAE = 0.42$ m, and $RMSE = 0.077$ m. These results show that ICESat-2 can be combined with remote sensing data to realize the long-time series dynamic monitoring of water bodies in plateau lakes, and it can be used as a supplement for no measured data.

(4) In addition, ICESat-2 shows great potential in accurately describing underwater topography and captures nearly 30 m of water depth signals reflected from underwater in the Ngoring Lake basin. In future research, the use of the ICESat-2 dataset and other satellite data with global spatial coverage (such as optical remote sensing or waveform data) will enable the long-term monitoring of changes in the overall water resources of the TP. This approach helps to better predict the temporal and spatial characteristics of the regional water level changes in the TP and provide a basis for hydrological simulation in the plateau region. It is greatly important for further research on the influencing factors of water level fluctuations and the formulation of reasonable water resource dispatching and management policies.

**Author Contributions:** Conceptualization, W.Z. and S.Z.; methodology, Z.Q.; software, S.Z.; validation, N.H., Y.L., Z.Z. and F.Y.; formal analysis, W.Z. and Y.L.; investigation, S.Z.; resources, S.Z.; data curation, S.Z.; writing—original draft preparation, S.Z. and W.Z.; writing—review and editing, S.Z.; visualization, S.Z.; supervision, W.Z.; project administration, W.Z.; funding acquisition, W.Z. All authors have read and agreed to the published version of the manuscript.

**Funding:** This work was supported by the National Key R&D Program of China (2016YFC1400904) and the scientific innovation program project by the Shanghai Committee of Science and Technology (20dz1206501).

**Data Availability Statement:** The data are available to readers by contacting the corresponding author.

**Acknowledgments:** We would like to acknowledge the NASA National Snow and Ice Data Center, European, LEGOSHYDROWEB for providing ICESat-2, Landsat series, Sentinel-2, and altimeter data, respectively. We appreciate the comments of the three anonymous reviewers and editors, which helped improve this study.

**Conflicts of Interest:** The authors declare that they have no known competing financial interests or personal relationships that could have appeared to influence the work reported in this paper.

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
