# Peer review of "Monitoring and Analysis of Water Level–Water Storage Capacity Changes in Ngoring Lake Based on Multisource Remote Sensing Data"

_water, doi:10.3390/w14142272_

Round 1
Reviewer 1 Report
The paper of Weidong Zhu et al. “Monitoring and Analysis of Water level–Water Storage Capacity Changes in Ngoring Lake based on Multisource Remote Sensing Data” is interesting and probably represents the first experience of the authors in such generalizations. However, I would like to emphasize in the Introduction the novelty of the current study in comparison with the studies presented in previously published works (References 47-50). Given that the authors promise "In the future, we will consider combining the actual bathymetric data of the lake and fusing different altimetric satellite data to deepen the research on water level and storage capacity", it is advisable to support them in this endeavor.
Minor
References
Page 21, Lines 558-559
You wrote:
ZHU, W.; YAN, J.; JIA, S., Lake status, major problems and protection strategy in China. Remote Sensing 2017, 9(10), 1032.
But I found such References:
Zhu Wenbin, Jiabao Yan, Shaofeng Jia. Monitoring Recent Fluctuations of the Southern Pool of Lake Chad Using Multiple Remote Sensing Data: Implications for Water Balance Analysis. Remote Sensing 2017, 9(10), 1032.
G Yang, R Ma, L Zhang, J Jiang, S Yao, M Zhang, HD Zeng. Lake status, major problems and protection strategy in China. Journal of Lake Sciences 2010, 22(6), 799-810
Also
Page 21, Line 581
You wrote:
Fpa, D.; Ahb, C.; Hcja, C. et al.
In fact:
Frederick Policelli, Alfred Hubbard, Hahn Chul Jung, Ben Zaitchik, Charles Ichoku
Please check these and all other References.
Page 14, Lines 326-328
You wrote: Combined with linear regression analysis, in general, the order of monthly average water level change rate from large to small is as follows: non-freezing period, freezing period, and melting period.
But Figure 8 shows, I think, that the order of monthly average water level change rate from large to small is as follows: non-freezing period, melting period and freezing period (blue - green - yellow).

Author Response
Dear reviewers, experts:
Greetings! First, thank you for your patience and careful guidance. In response to the review comments of the reviewer, I combined the original article to answer the questions and form the text. In order to facilitate the experts to re-examine, I will carry out the various problems pointed out by the experts. The following is a one-to-one answer to the experts' comments, and the key contents are marked in red (also marked in the revised paper). The blue mark in the paper is the language expression modification part of the paper.
Finally, thank you again and the reviewers for your valuable comments! And at the same time, I hope that if you find any deficiencies again during the review process, please inform us in time, and I will make further amendments to the suggestions.
Sincerely,
Zhao. shubing

Reviewer 2 Report
Title
The Title reflects the paper’s content accurately.
Abstract
The Abstract determines the paper’s content and objectives in a very manifest and complete fashion.
1. Introduction
Start off the Introduction by “Water is a finite, vulnerable and essential resource which should be managed in an integrated manner.” [1] and then go on to lakes. Also in L32, it should be mentioned that extreme precipitation [2] plays a role in the Tibetan Plateau as seen in [3], [4],[5]. Otherwise, the Introduction is both adequate and highly informative.
2. Materials and Methods
Includes accuracy verification and evaluation of ICESAT-2 and satellite altimeter. Well-described.
3. Results
In L277 add ‘ which has different response than alpine meadows [6]’. Quite exhaustive and to the point.
4. Discussion
Extensive and detailed.
5. Conclusions
Precise and firmly based on the previous sections.
References
[1] Zisopoulou, K., D. Zisopoulos, and D. Panagoulia, “Water Economics : An In-Depth Analysis of the Connection of Blue Water with Some Primary Level Aspects of Economic Theory I,” Water (Switzerland), vol. 14, 2022, doi: https://doi.org/10.3390/w14010103.
[2] Panagoulia, D. and E. I. Vlahogianni, “Nonlinear dynamics and recurrence analysis of extreme precipitation for observed and general circulation model generated climates,” Hydrol. Process., vol. 28, no. 4, pp. 2281–2292, 2014, doi: 10.1002/hyp.9802.
[3] Gao, J., P. Ma, J. Du, and X. Huang, “Spatial distribution of extreme precipitation in the Tibetan Plateau and effects of external forcing factors based on Generalized Pareto Distribution,” Water Sci. Technol. Water Supply, vol. 21, no. 3, pp. 1253–1262, 2021, doi: 10.2166/ws.2020.365.
[4] Wang, W., H. Li, Z. Xie, X. Zhu, L. Xiao, X. Hao, and J. Wang, “Continental Water Vapor Dominantly Impacts Precipitation during the Snow Season on the Northeastern Tibetan Plateau,” J. Clim., vol. 35, no. 12, pp. 3819–3831, 2022, doi: https://doi.org/10.1175/JCLI-D-21-0241.1.
[5] Sun, J., X. Yao, G. Deng, and Y. Liu, “Characteristics and synoptic patterns of regional extreme rainfall over the central and eastern tibetan plateau in boreal summer,” Atmosphere (Basel)., vol. 12, no. 3, 2021, doi: 10.3390/atmos12030379.
[6] Hao, A., H. Duan, X. Wang, G. Zhao, Q. You, F. Peng, H. Du, F. Liu, C. Li, C. Lai, et al., “Different response of alpine meadow and alpine steppe to climatic and anthropogenic disturbance on the Qinghai-Tibetan Plateau,” Glob. Ecol. Conserv., vol. 27, no. April, p. e01512, 2021, doi: 10.1016/j.gecco.2021.e01512.
Author Response

(The authors gave the same response as above.)

Round 2
Reviewer 1 Report
The authors have addressed all my comments to great extent
I recommend paper publication

This manuscript is a resubmission of an earlier submission. The following is a list of the peer review reports and author responses from that submission.